# Sinusoidal Control of a Brushless DC Motor with Misalignment of Hall Sensors

**Krzysztof Kolano** [1,*], **Bartosz Drzymała** [1] **and Jakub Gęca** [2]

1 Faculty of Electric Drives and Machines, Lublin University of Technology, Nadbystrzycka 38a, 20-618 Lublin, Poland; b.drzymala@pollub.pl
2 Doctoral School, Lublin University of Technology, Nadbystrzycka 38a, 20-618 Lublin, Poland; j.geca@pollub.pl
* Correspondence: k.kolano@pollub.pl

**Abstract:** This article presents an estimation method of the BLDC rotor position with asymmetrically arranged Hall sensors. Position estimation is necessary to control the motor by methods other than block commutation. A sinusoidal control method was selected for the research, which significantly reduces torque ripples and acoustic noise and is quite simple to implement. Inaccurate performance of the elements determining the position of the BLDC motor rotor causes a large error in the position estimation and has a negative impact on the operation of the drive controlled in this way. Using the developed control algorithms, it is possible to correctly determine the mechanical position of the rotor even for multi-pole motors. The proposed method is relatively easy to implement and does not require modification of control systems, being limited to changes only in the software of such devices. The tests of the actual system clearly show the usefulness of such a control method and its effectiveness.

**Keywords:** rotor position estimation; BLDC motor; sensor misalignment





## 1. Introduction

Brushless motors are gaining more and more popularity due to their operational properties and higher energy efficiency compared to DC motors. Especially in battery devices, where the amount of useful energy is limited, the advantages of these machines in comparison to DC motors are noticeable for the user [1,2].

Their unquestionable disadvantage is the complicated electronic system, which functionally corresponds to advanced full-bridge power converters and the necessity to use elements that determine the position of the device rotor [3]. These components include, but are not limited to, encoders, resolvers, Hall effect sensors [4] and image sensors [5]. For financial reasons, among the elements that determine the rotor position, solutions based on Hall sensors are the most common in BLDC machines. Therefore, continuous attempts have been made to improve the performance of BLDC motors with Hall sensors [6], which also has the effect of reducing the torque pulsations of these devices [7]. Moreover, a number of vibration analyses of BLDC motors for diagnostic purposes can be seen in the literature [8,9]. Additionally, the block commutation control method contributes to the formation of undesirable acoustic effects related to the commutation processes of the BLDC motor windings [10].

In order to minimize the increased noise, a sinusoidal BLDC motor control method has been developed, which is some form of intermediate between field-oriented vector control and block control of the motor. As a relatively simple method, it does not require the construction of systems for accurate measurement of the motor phase currents, but only an algorithm that estimates the position of the BLDC motor rotor based on the change in the state of the Hall sensors determining the position of the shaft. This method provides good results in steady-state operation of such a machine while reducing the emitted noise [11,12].

The position is estimated by calculating the rotor's position using the known speed of its shaft and information about the time that has elapsed since the last computation of its position [13]. It becomes clear that the basis for the correct determination of the rotor position is the correct reading of the machine rotational speed and the correct determination of the rotor position at the time of the change in the Hall sensor state [14]. In a BLDC motor with shaft position sensors, the simplest and, at the same time, cheapest method of determining the shaft rotational speed is to calculate the time between the change in the rotor position sensors' state. The quotient of the machine design constant taking into account the number of pole pairs and the time of change in the commutator sensors' state allows determining the rotational speed in rpm.

Such assumptions work very well in systems in which the rotational speed of the motor shaft can be determined precisely on the basis of the changing Hall sensors' state. The accuracy of this measurement is crucial in the process of estimating the position of the motor shaft. In practice, many researchers point to the accuracy of the placement of the motor shaft position sensors [15–17], which depends on the accuracy of the production process, which significantly affects the value of the speed calculated in this way. An incorrect speed value leads to an incorrect motor position estimation, which in turn causes the modulator to generate an incorrect control voltage vector.

This paper presents a method for estimating the rotor position of a BLDC motor with Hall sensor misalignment. The method analyzes the error distribution resulting from the inaccuracies of the rotor position determination components and then introduces a commutation correction process. This method was successfully implemented for the sinusoidal control of the BLDC motor, which made it possible to avoid the necessity of the incremental encoder usage. The obtained results show that the developed control method with commutation correction allows for a significant reduction in torque fluctuations on the machine shaft, thus reducing the level of acoustic noise.

## 2. Position Estimation

Information about the rotor position of a BLDC motor equipped with Hall sensors is not continuous and is limited to the moment of changes in the state of the sensors. In the next steps, an estimation of the position is made on the basis of the calculated rotational speed of the rotor and the time.

Each change in the Hall sensors' state is a signal for the estimation algorithm to increase the angular position of the rotor by the value of the mechanical angle $\Theta_{sec}$. Each time in such a situation, a procedure is performed to synchronize the value of the calculated position with the actual value (Figure 1). This procedure is especially important in dynamic states where the estimation action may be burdened with an error related to, for example, a step change in the load torque. Additionally, each time in such a situation, the current angular velocity of the shaft is calculated, which will be used to calculate the position in the next sector of the rotor position angle:

$$\Theta_{Mn} = f(H1, H2, H3, \omega_{n-1}, \Delta t) \tag{1}$$

The angular position $\Theta_{Mn}$ of the rotor of a brushless DC motor equipped with shaft position sensors can be calculated for n-sector from the formula

$$\Theta_{Mn} = (n-1) \cdot \Theta_{sec} + \Delta\Theta_{Mn} \quad \Delta\Theta_{Mn} = f(\omega_{n-1}, \Delta t) \tag{2}$$

It is worth emphasizing that the estimated change in the angle $\Delta\Theta_{Mn}$ is influenced by the rotational speed measured in the previous sector of the Hall sensors' state, and its correct value is of key importance in this process.

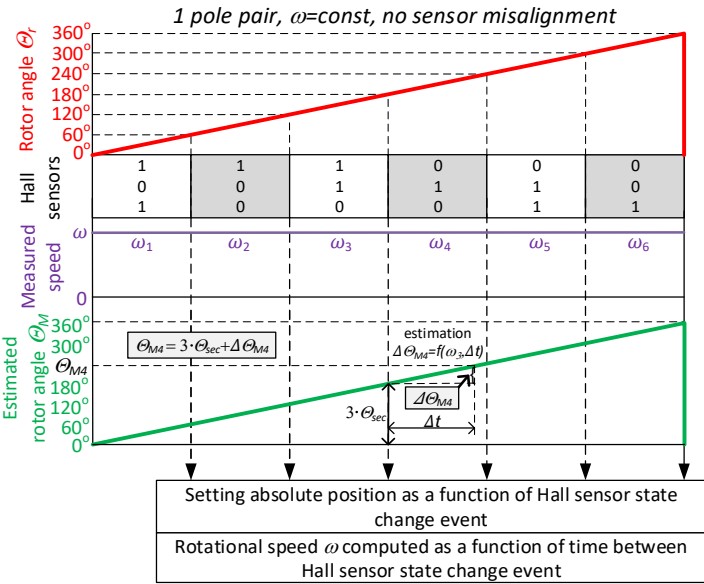

**Figure 1.** Estimation of the angular position of the BLDC motor based on the change in the Hall sensors' state for the correct arrangement of the sensors.

### 3. Errors in the Placement of Sensors

In order to verify the commonness of errors in the arrangement of sensors, it was decided to purchase 30 BLDC motors from 1 production batch so that they could be compared reliably. In order to determine the inaccuracy of the construction of the motor shaft position observation system, the software was prepared in accordance with the algorithm presented in [18]. Each change in the Hall sensors' state causes the readout of the change in the motor shaft position by the mechanical angle:

$$\Theta_{sec} = \frac{360^{\circ}}{6 \cdot p} \tag{3}$$

Which, for a typical four-pole motor, is 15 mechanical degrees.

The asymmetric arrangement of the Hall sensors leads to their activation at the wrong moment, which translates into an inaccurate reading of the mechanical position of the motor rotor.

The difference $\Theta_{esec}$ between the real value of mechanical degrees between successive changes in the Hall sensors' state and the value of $\Theta_{sec}$, is directly correlated with the error in the implementation of the shaft position observation system. The average error $e_{avg}$ is the average difference in determining the position of the motor shaft based on the change in the state of the sensors according to the formula

$$e_{avg} = \frac{100\%}{\Theta_{sec}} \cdot \sum_{i=1}^{i=6 \cdot p} |\Theta_{esec}| \tag{4}$$

and is expressed as a percentage.

Figure 2 below shows the mean percentage error $e_{avg}$ and the recorded maximum error $e_{max}$ for each of the tested motors.

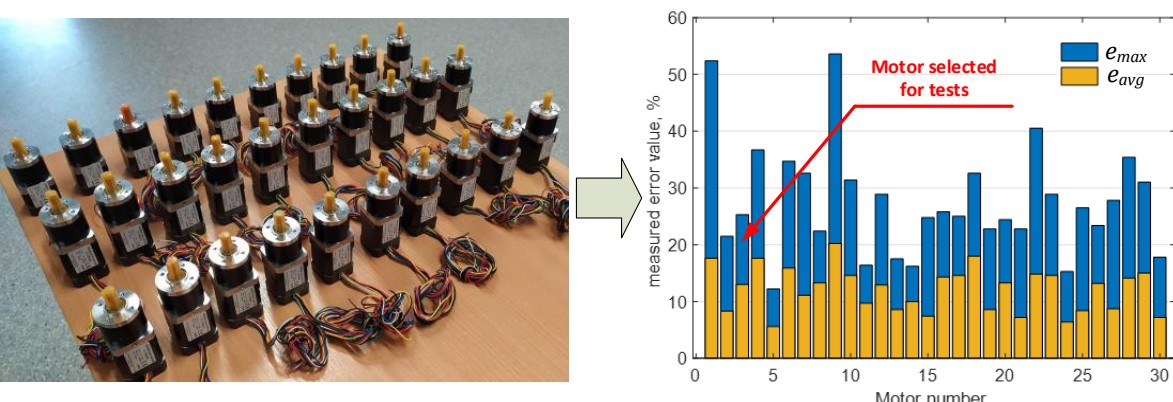

**Figure 2.** Distribution of errors in the position of sensors for BLDC motors from one production series.

It is clearly seen that this problem is common, and that the sensors' placement error value is significant with respect to the absolute values of the motor shaft position determination. This results in an erroneous determination of the rotor position at the moment of the Hall sensors' state change, as well as high inaccuracy in determining the motor rotational speed on the basis of the time measured between successive changes in the Hall sensors' state (Figure 3) [18].

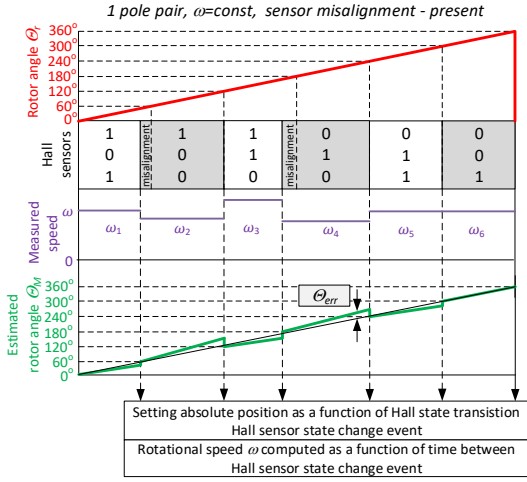

**Figure 3.** Estimation of the angular position of the BLDC motor based on the change in the Hall sensors' state for the actual arrangement of the sensors.

Due to the inaccurate arrangement of the shaft position sensors, both the rotor's position during the change in the sensors' state and its rotational speed are incorrectly determined. This has a key impact on the process of estimating the angle of the motor shaft. These factors cause a cumulative error in the estimation of the shaft position $\Theta_{err}$, which significantly worsens the quality of the motor's sinusoidal control. The situation is even worse when a multipolar motor is controlled. Since the estimated electric angle is a value $p$ times greater than the value of the estimated angle of the shaft mechanical position ($p$—number of pole pairs), the value of the estimation error also increases $p$ times. Since most BLDC motors are multipolar machines (usually $p = 2$ or $p = 4$), the error in estimating the electric angle $\Theta_{e\_err}$ takes significant values:

$$\Theta_{e\_err} = p \cdot \Theta_{err} \tag{5}$$

An incorrectly determined angle causes the control system to select an incorrect motor control vector, causing a fluctuation in the driving torque, which, in an extreme case, may prevent the drive from working.

## 4. Laboratory Stand

During the work on the estimation algorithm taking into account sensor placement errors, it was necessary to analyze many electrical and mechanical quantities, both from Hall sensors themselves and the estimated value of the rotor angle and speed measured by the control microcontroller. Due to the strict temporal correlation of the electrical quantities and the values calculated by the microprocessor system, it was decided to use a unit equipped with two digital-to-analog (DA) converter channels—the 32-bit STM32F303 microcontroller supported by STMicroelectronics N.V. (Amsterdam, The Netherland). In addition, the absolute rotor position was determined using the AS5047n system—a 14-bit absolute magnetic encoder supported by amsAG (Premstaetten, Austria) and the STM32F072 microcontroller, whose task was to convert the numerical value of the position into a voltage signal (Figure 4). All the electrical signals were connected to and supported by the Tektronix (Beaverton, OR, USA) Mixed Signal Oscilloscope MSO 464-BW-1500.

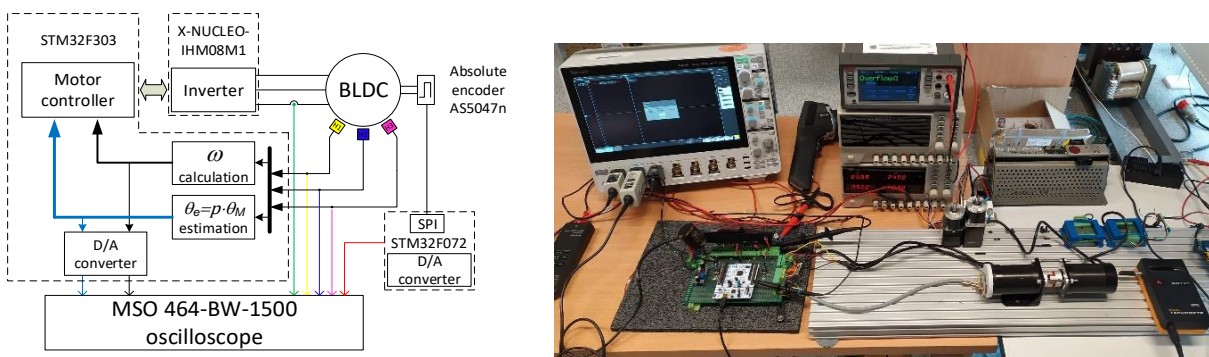

**Figure 4.** BLDC motor test stand for various control methods ($\omega$—calculated as a function of time measured between the successive changes in the Hall sensors).

During the steady-state operation of the system in an open feedback loop, the waveforms of signals from Hall sensors, the motor shaft speed $\omega$, the estimated electric angle $\theta_e$ and the motor phase current were recorded (Figure 5).

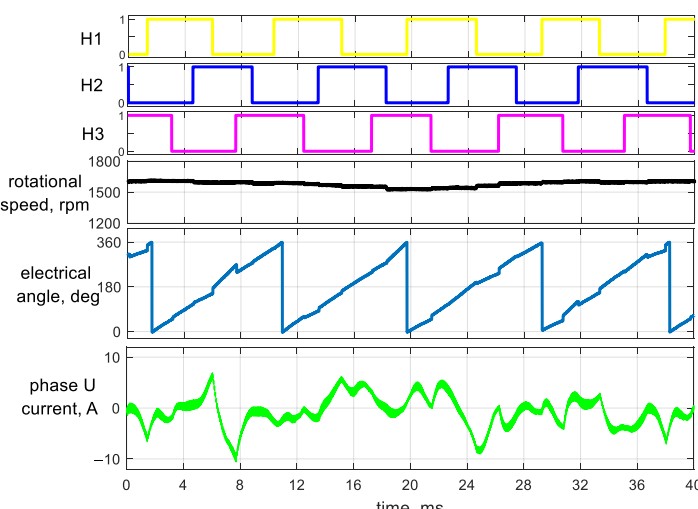

**Figure 5.** Waveforms recorded using the sinusoidal control technique of the BLDC motor with classic rotor position estimation.

It can be clearly noticed that the estimated value of the electric angle is strongly non-linear. This results directly from the error related to the sensor misalignment. At the same time, the erroneous generation of the voltage vector significantly affects the wave of the motor phase current. Incorrect determination of the rotor position angle has a large

impact on the generated electromagnetic torque, which translates into a noticeable speed fluctuation in the steady state. In addition, the noise emitted by the motor in relation to the block control did not decrease at all. Moreover, during its operation, significant vibrations could be noticed and felt. These phenomena make the use of this type of control often ineffective and worsen the operation of the BLDC drive system.

## 5. Example Implementation

As part of the work related to sinusoidal control, it was found that all tested motors, equipped with shaft position sensors, had non-zero placement errors. It is therefore desirable to develop a method that is insensitive to such imperfections of these systems.

The error in determining the angular position consists of two factors:

(a) A factor directly related to the inaccuracy of sensor installation—when their state changes, the algorithm incorrectly assumes their precise arrangement and assigns an incorrect value according to the formula $\Theta_{Mn} = (n-1) \cdot \Theta_{sec}$;

(b) Incorrect placement of sensors affects the speed determination error, which in turn causes position estimation errors in the time between the change in the sensors' state.

Speed reading correction algorithms have been developed in many publications [13,15,17,19] and are successfully used, significantly contributing to the improvement in BLDC motor operation in a closed feedback loop. This article describes how to determine the actual rotor position of a BLDC motor when the state of the rotor position observation outputs changes.

With block control in the steady-state operation of the BLDC motor, the rotor speed changes very slightly; therefore, one can assume a linear increment of the rotor angle per unit of time—Figure 1. If, at the moment when the state of the sensors changes to an angular position, the microcontroller timer is started, then the course of the value incremented by a constant register time interval will be analogous to the course of the angular position of the rotor (Figure 6).

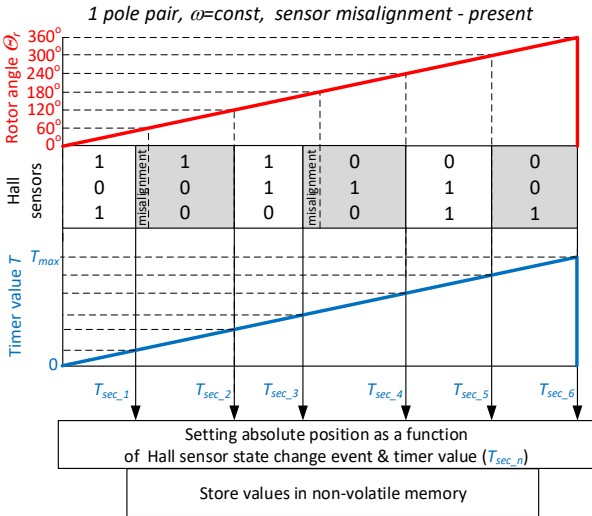

**Figure 6.** The proposed method of estimating the angular position of the BLDC motor based on the change in the Hall sensors' state for the actual arrangement of the sensors.

By saving the state of the timer register during successive changes in the sensors' state ($T_{sec\_1} do T_{sec\_6}$), it is possible to calculate the real angle corresponding to the successive switching of the shaft position sensor signals according to the formula

$$\Theta_{Mn} = 360° \cdot \frac{T_{sec\_n}}{T_{sec\_6}} \qquad (6)$$

From now on, each change in the sensors' status will be a signal for the control system to update the rotor position to the value that was recorded during this process—Figure 7. When the motor is operating with an inaccurate arrangement of sensors, this position will not be burdened with an error resulting from this fact. It is worth noting that this is a one-time procedure, and the measurement results of the actual rotor position angle at which the state of the sensors changes are saved in non-volatile memory.

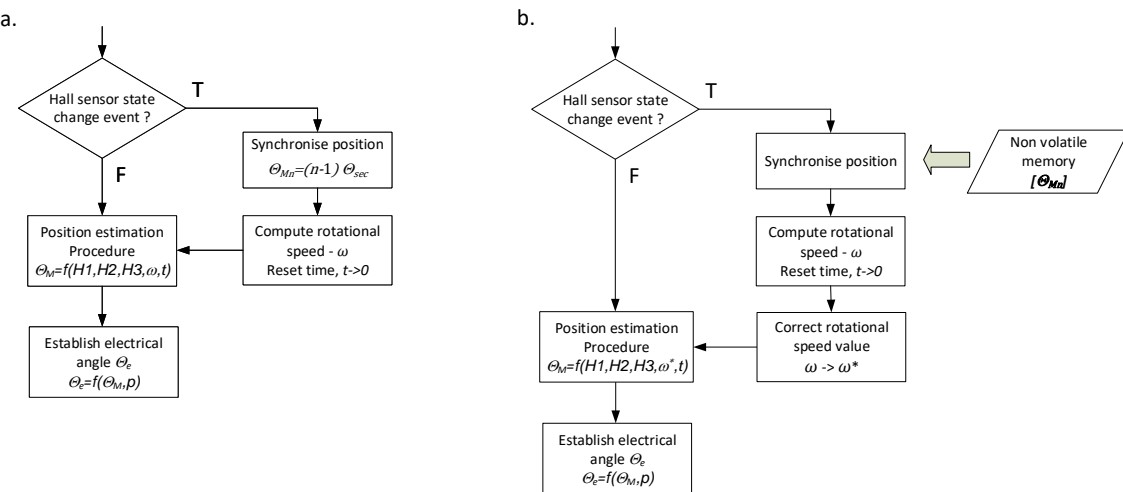

**Figure 7.** Classic position estimation algorithm based on the analysis of changes in the Hall sensors' state (**a**), and the proposed control algorithm (**b**).

Assuming the existence of errors resulting only from improper installation of sensors, and not taking into account errors resulting from inaccuracies in the execution of the rotating magnetic element, recording six values of the angle $\Theta_{Mn}$ is enough to correctly determine the position when the state of the sensors is changing. Unfortunately, these errors are most often correlated with those that result from inaccuracies in the execution of the rotating element on the motor shaft [18] (usually magnetic). This makes it necessary for this method to function properly to record six $p$ values corresponding to the respective angular position of the rotor. In multipolar BLDC motors, it is necessary to use the method of determining the actual, initial position of the motor, which is impossible when analyzing only the Hall sensors' state (reading the state of the sensors only allows determining the electric angle $\Theta_e$). Such a method has been developed and patented [20] and consists in identifying the position of the rotor on the basis of the distribution of speed errors for individual sectors [18].

Determining the actual rotor position allows the use of the developed correction method in the case of multipolar motors. This article focuses on the most complex problem concerning a motor with $p = 4$.

## 6. Research

To verify the theoretical assumptions, sinusoidal controlled BLDC motor driver software was prepared.

The tests were carried out for the NEMA 23 BLDC motor marked in Figure 2 with the following data:

| | |
|---|---|
| Nominal power | 60 W |
| Nominal speed | 3000 rpm |
| Nominal voltage | 24 Vdc |
| Pole pairs | 4 |

The tests were carried out for a motor started by the block-PWM method. After the motor obtained a steady speed, errors in the shaft position determination elements were determined. These errors are related to both the Hall sensors' location errors and the inaccuracy in the execution of the rotating magnet on the motor shaft [19]. Then, the modulation method was changed from block commutation to sinusoidal, and it was carried out with the corrected rotor position values assigned to the Hall sensors' switching points and the corrected values of the calculated motor rotational speed according to the algorithm [21]. The measurement results are shown in Figure 8.

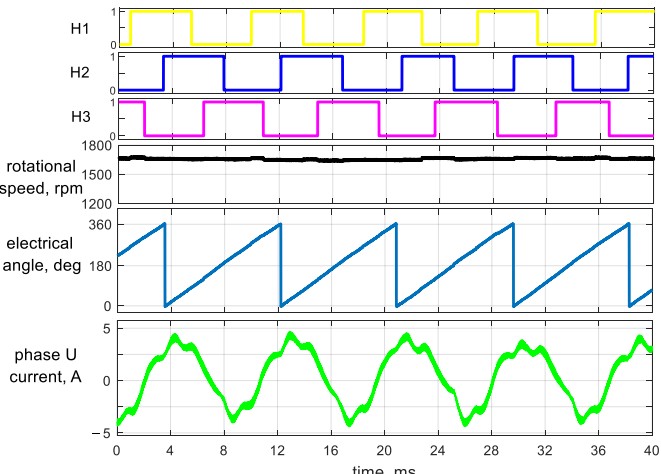

**Figure 8.** Waveforms recorded using the sinusoidal control technique of the BLDC motor with the proposed method of rotor position estimation.

Since the test object was a motor with *p* = 4, the number of points at which the state of the sensors changes is 24 per 1 full mechanical rotation. This means that for one full mechanical revolution, there are four complete periods of voltage supplying the motor. Compared to the motor measurements carried out for the standard control algorithm, it is clearly visible that the current waveform is much more similar to a sinusoid, and the waveform showing the change in the estimated electric angle is correct and reflects its actual change, despite the presence of significant errors in the arrangement of the elements determining the position of the motor rotor. Additionally, an almost unchanged plot course of the motor rotational speed is visible—it is related to smaller fluctuations of the electromagnetic torque (Figure 8).

The error in determining the electric angle translates directly into the achievable electromagnetic moment of the motor. The course of the maximum electromagnetic torque as a function of the mechanical rotor position angle for both sinusoidal control methods is shown in Figure 9. It is clearly visible that for the standard rotor position estimation procedure, the value of the maximum torque is strongly related to the accuracy of the components of the shaft position observation system and is reduced by 15% compared to the system with a correctly determined rotor position. For the proposed algorithm, the decrease in the maximum torque reachable by the motor in the examined case is only about 2%.

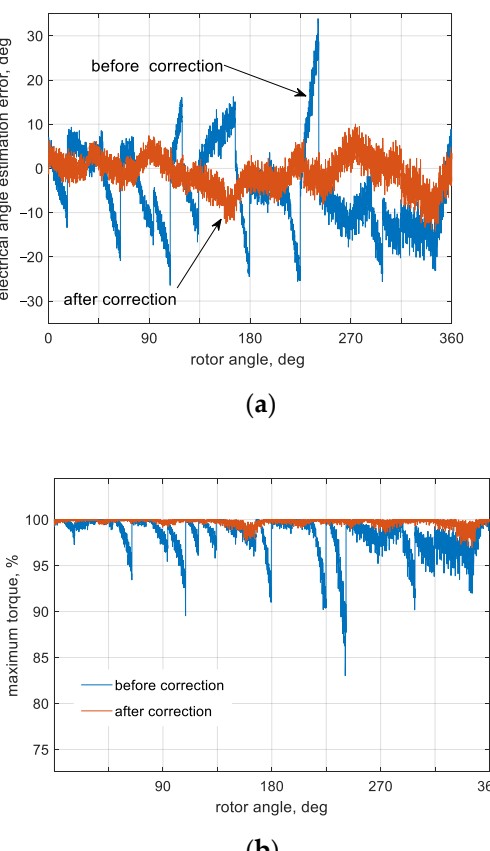

(**a**)

(**b**)

**Figure 9.** The course of the error in electric angle estimation as a function of the rotor position of the tested BLDC motor (**a**), and the course of the maximum possible motor torque (**b**). Standard estimation method—blue; proposed method—red.

In order to confirm the usefulness of the developed algorithm, it was decided to carry out transient measurements of the drive system. The motor with the highest inaccuracy of the shaft position observation system was selected for the tests. The mechanical rotor angle estimation process and the activation moment of the proposed method are shown in Figure 10.

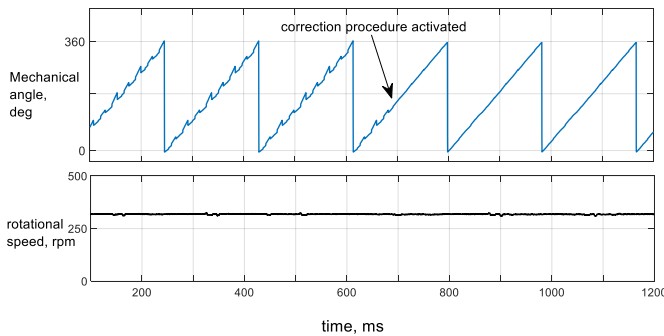

**Figure 10.** The course of the estimated $\Theta_M$ angle and the measured rotational speed with the moment of starting the proposed estimation procedure marked.

During the tests, the load of the shaft was realized by the same machine coupled on a common shaft, the windings of which were connected to a variable resistance load.

The waveforms of the corrected rotational speed measured using Hall sensors, the motor phase U current and the electric angle waveform were recorded. Figure 11 shows

these waveforms for a step load change corresponding to 60% of the nominal torque for a system operating in the open-loop speed control.

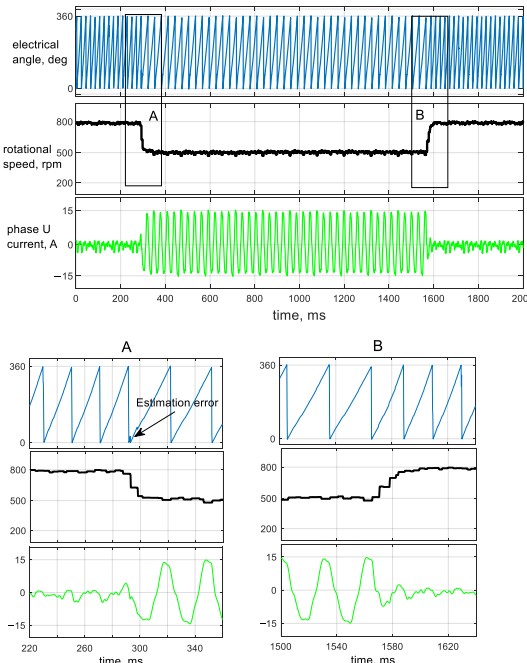

**Figure 11.** The course of the estimated value of the $\Theta_M$ angle, the measured rotational speed and the motor phase U current, for the system operating in the open-speed loop regulation system for a load torque change of 0–60–0% of nominal torque.

The measurements were repeated for the closed-loop speed regulator system, and the results are presented in Figure 12.

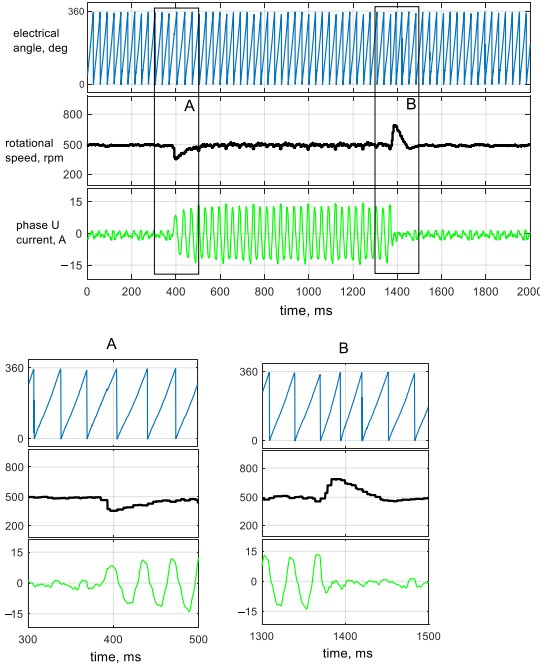

**Figure 12.** The course of the estimated value of the $\Theta_M$ angle, the measured rotational speed and the motor phase U current for a closed-speed loop regulator system and a load torque change of 0–50–0% of nominal torque.

What is worth emphasizing is that the input signal of the speed controller is not the reference value measured by the AS5047n sensor, but the value measured on the basis of changes in the state of the Hall sensors and corrected by the procedure proposed in [21].

The next graph (Figure 13) shows the response of the drive system to a sudden change in the speed reference for closed-loop speed operation.

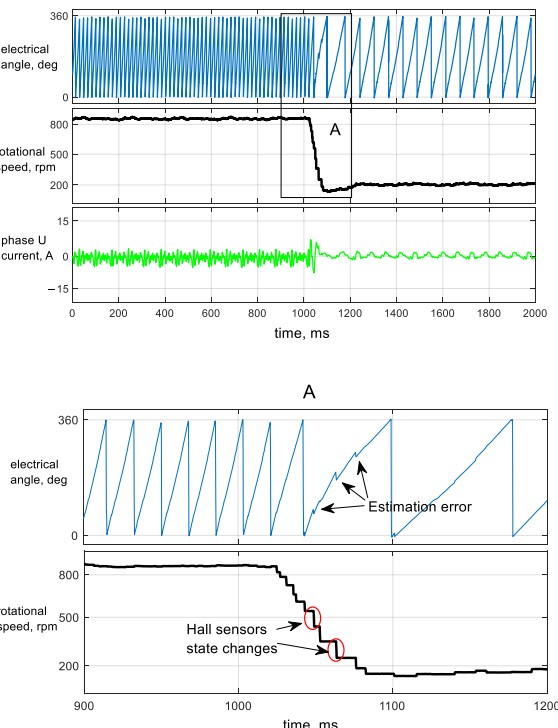

**Figure 13.** Waveform of the estimated value of the $\Theta_M$ angle, the measured rotational speed and the motor phase U current for the closed-speed loop regulator system and changing the setpoint from 900 to 500 rpm for idle state.

As the transient speed value decreases, the position estimation error increases until the Hall sensors change state. A change in the state of the sensors is a signal to update the actual position of the motor shaft. This error is clearly visible in Figure 13A, but also in this case, the drive worked properly. During the investigations of transient states, no abnormalities in the proposed control method were found. Additionally, due to the fact that only changes in the state of the Hall sensors are analyzed, the method is not dependent on the electrical parameters of the motor.

## 7. Conclusions

The use of the sinusoidal control of the BLDC motor is intended to reduce the fluctuations in the motor electromagnetic torque, which significantly reduces the level of acoustic noise. An additional advantage is the ability to control motors with sinusoidal induction distribution, equipped with a simple rotor position determination system based on a rotating magnet and a set of three Hall sensors.

Unfortunately, the rotor position-determining components are usually made with limited accuracy, which negatively affects both the accuracy related to the position determination of the rotor and the calculation of its rotational speed. Both of these values are key in the shaft position estimation process and thus deteriorate the accuracy of generating the appropriate power parameters.

This article presents a new method of BLDC rotor position estimation with shaft position sensors. This method works well in motors where the shaft positioning system is not made with high accuracy. The proposed method is also suitable for motors with

many pole pairs, which require the application of the mechanical position determination procedure using the analysis of the error distribution resulting from the accuracy of the rotor position determination system. The use of this method allows applying sinusoidal control without the need to install an incremental encoder in motors with significant construction inaccuracies, which makes it easier to use and reduces investment costs.

**Author Contributions:** Conceptualization, K.K.; methodology, K.K.; software, J.G.; validation, K.K., J.G. and B.D.; formal analysis, K.K.; investigation, B.D.; resources, K.K. and J.G.; data curation, B.D.; writing—original draft preparation, K.K.; writing—review and editing, K.K. and B.D.; visualization, J.G.; supervision, K.K.; project administration, K.K.; funding acquisition, K.K. All authors have read and agreed to the published version of the manuscript.

**Funding:** This work was supported by Lublin University of Technology grant no. FD-20/EE-2/605.

**Institutional Review Board Statement:** Not applicable.

**Informed Consent Statement:** Not applicable.

**Conflicts of Interest:** The authors declare no conflict of interest.

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
