# Peer review of "Sinusoidal Control of a Brushless DC Motor with Misalignment of Hall Sensors"

_energies, doi:10.3390/en14133845_

Round 1
Reviewer 1 Report
Dear Author,
1) Please avoid the lumped reference. The introduction section need to improve with propoer citations. The introduction is too short and need to add main contribution of the article.
2) Section 2 and 3 provides a general information. Please avoid well known theroy and present only new importat points.
3) The experiemental setup is given. However, the experimental results are not given.
4) Please provide dyanamics of the system under various conditions.
5) Please add complete control and algorithm detail of the system.
6) The detail proposed work need to present.
Author Response
I revised manuscript in accordance to Your suggestions.
Reviewer 2 Report
Authors are presentting novell compensation method for BLDC with sinusoidal control. I am missing more explanations of the strategy advantages in the paper. The method is proved only by last figure. To increase scientifict impact of the paper I recomed to insert more detailed analysis of the influence of sensors inaccuracy on BLDC motor torque under sinusoidal supply. Explain better your proposed strategy and support it with more measurement like (different speeds, different load, speed transients etc.) Questions and comments to the authors:
1) In stable operational state the method should work even with one hall sensor. Where the angle will increase linearly, where is your improvement?
2) Explain meaning of thetaMn^2 in eq 2.
3)Are fig. 5 and fig. 8 recorded under same conditions? Explain source of current distortion when sinusoidal control is assumed.
4) I would suppose that electrical angle in fig 5 will have 4 times same shape caused by hall sensor misalignmet. Eplain why not.
5) In chapter 6 you are using word engine instead of motor.
6) Explain better the test described in chapter 6. There is sentence "Then the modulation method was changed to sinusoidal and ..." What was the modulation method before. How was the motor controlled?
Author Response
Ad 1.
When the state of the hall sensors changes, the mechanical position of the rotor is updated immediately (incrementation/decrementation). In motors where the Hall sensors switch at the wrong time, the updated position is incorrect and a position determination error is generated, and the estimation is done for incorrect initial angle position and incorrect rotational speed value.Determining the position and rotational speed from the state changes of one sensor during steady operation would be possible, but the frequency of the actual motor shaft position readings would be low and the dynamics of operation would be limited. The proposed method makes it possible to determine the actual position of the motor shaft for each change in the state of the hall sensors - which ensures a much higher frequency than the use of one sensor. In addition, the determination of the actual speed (regardless of the accuracy of the placement of sensors) contributes to the improvement of the accuracy of position estimation between changes in the states of Hall sensors.
Ad 2.
This was a formatting glitch.
Ad 3.
The conditions were the same – low motor load (less than 10%), and sinusoidal control active. Because of the idle state of the motor current is only quasi-sinusoidal. Additionally closed loop speed regulator was active and could affect current shape by modyfying PWM register values.
Ad 4.
According to my researches, hall sensors misalignment is always present at the low-end BLDC motors. It is always (as far as I tested) combined with limited accuracy of rotating magnet. Both this errors combine into unique shape 1 time per mechanical revolution – each plot of electrical angle for the full mechanical revolution is unique. (Such sensor, supported by additional procedure can be treated as an absolute encoder).
Whole investigation has been presented in: Kolano, K. Determining the Position of the Brushless DC Motor Rotor. Energies 2020, 13, 1607. https://doi.org/10.3390/en13071607
Ad 5.
Of course – translators mistake.
Ad 6.
The tests were carried out for an motor started by the block- PWM method (trapezoidal cotroll).
Reviewer 3 Report
Overall, a well-written paper that offers a new method for estimating the rotor position of a BLDC. However, some of the test results presented in the paper seem too ideal and may not reflect real-world scenarios.
- In line 182, it is assumed that the angular error results from the improper installation of sensors. Still, in reality, heating and physical impact could affect the accuracy of angular measurement. How does the proper scheme correlate with the real-world scenario of machines?
- There is no consideration for transient operation, including considerable variation in load and operating speed. How does the proposed method behave for a low, medium and full speed range of operation? Perhaps, more test is required to validate this for various nature of loads.
- The method proposed in Figure 7 relies on non-volatile memory for storing the static state of the sensors; perhaps a more adaptive approach like the artificial neural network adaptive referencing method (implemented as a feedback loop) could solve this issue and hence minimise unforeseen errors.
Author Response
Ad 1.
During the tests, we focused on machines with sensors located outside the motor housing, which makes it easier to neglect temperature changes. However, it is possible to cyclically determine new sensor placement errors each time the motor is operating at steady state. This enables the actual angles and values of rotational speeds to be corrected.
Ad. 2
I revised the manuscript in accordance to Your suggestions.
Ad 3.
This is very interesting idea. I will try those method in the future work.
Round 2
Reviewer 1 Report
Dear Author,
Please add results of experimental work with full screen of scope.
The dynamics of the system need to investigate.
Author Response
First of all, thank You for Your valuable comments on the article. To meet the requirements drawings of transients have been added.Image files from the oscilloscope are also included as additional material.
Reviewer 2 Report
The required experimets showing the behavior of the controlled BLDC under dynamic operation have been presented. However I would like to see also some details.
English in added part is horrible (mostly word order). For example Fig. 12 - ... and the U motor current phase
Author Response
First of all, thank you for your valuable comments on the article. To meet the requirements Drawings of transients have been added.Image files from the oscilloscope are also included as additional material.
I also improved the English language in the added part of the article.
Reviewer 3 Report
The updated manuscript is in a much-improved form for publication. A minor recommendation is for the authors to either provide the equation used for calculating omega in Figure 4 or provide a reference.
There are still a few minor grammatical errors that require attention.
Author Response
Referring to the revision, I added a comment to Figure 4 regarding the method ofcalculating the motor rotational speed.
Round 3
Reviewer 1 Report
Thank you for your response. However, full screen Oscilloscope results are still missing.